# In Vitro Gastrointestinal Bioaccessibility, Bioactivities and Colonic Fermentation of Phenolic Compounds in Different *Vigna* Beans

**DOI:** 10.3390/foods11233884

**Published:** 2022-12-01

**Authors:** Minhao Li, Qian Bai, Jiajing Zhou, Thaiza Serrano Pinheiro de Souza, Hafiz Ansar Rasul Suleria

**Affiliations:** School of Agriculture and Food, Faculty of Veterinary and Agricultural Sciences, The University of Melbourne, Melbourne, VIC 3010, Australia

**Keywords:** pulses, phenolic compounds, digestibility, fermentation, antioxidant capacity, bioavailability, SCFAs

## Abstract

Beans are widely consumed throughout the world, rich in non-nutrient phenolic compounds and other bioactive constituents, including alkaloids, lectins, and others. However, research about in vitro digestion impacts on the changes of bioactive compounds’ release and related antioxidant potential in different *Vigna* beans is limited. This research aimed to assess the modifications that occur in the content and bioaccessibility of phenolic compounds in four *Vigna* samples (adzuki bean, black urid whole, black eye bean, and mung bean), their antioxidant properties, and short chain fatty acids (SCFAs) production through static in vitro gastrointestinal digestion and colonic fermentation. Adzuki bean exhibited relatively higher total phenolic content (TPC; 4.76 mg GAE/g) and antioxidant activities after in vitro digestion. The black eye beans’ total flavonoid content (0.74 mg QE/g) and total condensed tannins (10.43 mg CE/g) displayed higher tendencies. For colonic fermentation, the greatest TPC value of entire samples was detected through a 2-h reaction. In most selected beans, phenolic compounds were comparably more bioaccessible during the oral phase. Acetic acid showed the highest level through SCFAs production, and the total SCFAs in adzuki beans was the greatest (0.021 mmol/L) after 16-h fermentation. Adzuki beans may be more beneficial to gut health and possess a stronger antioxidant potential after consumption.

## 1. Introduction

The *Vigna* Fabaceae, closely correlated to *Phaseolus*, is one of the dominant genera in the pulse family, composing more than 200 species with a pantropical distribution [1]. Adzuki bean (*V. angularis*), mung bean (*V. radiata*), black urid whole (*V. mungo*), and black eye bean (*V. unguiculata*) are typical *Vigna* pulses that are widely cropped and consumed in Asia and Africa [2]. Adzuki beans are a popular Asian ingredient used in a wide range of dishes, including desserts, cakes, and porridge [3]. In India, Southeast Asia, and Western countries, mung bean seeds and sprouts are broadly used as fresh salad vegetables or typical food. *V. mungo* is commonly used in Northern India to make *dal* from whole or split, dehusked seeds [2]. Furthermore, black eyed beans are a type of legume that can be eaten both fresh and dried and nowadays are also grown in Europe, North, Central, and South America, the Caribbean, and Australia as well [4].

Rich in non-nutrient bioactive compounds such as phenolics, regular ingestion of *Vigna* species has been found with potential health benefits. For example, adzuki bean, showing high flavonoids, has been found with antibacterial as well as anti-diabetic potential [3,5]. Black eye bean has significant anti-inflammatory impacts and potential therapeutic advantages for diabetes, cancer along with cardiovascular disease, according to Winham et al. [6]. In addition, previous studies have demonstrated that black urid whole is abundant in several phenolic acids containing gallic, vanillic, caffeic, protocatechuic, syringic, and ferulic acid, favorable for the human body [4]. As for mung bean, catechin, quercetin, chlorogenic acid, caffeic acid, *p*-coumaric acid as well as kaempferol were illustrated. Enriching flavonoids in the seed and seed coat of mung bean, preventive functions on oxidative damage resulting from heart stress are displayed [7].

The health potential of bioactive phenolics is not only reliant on the amount ingested but also on their bioaccessibility [8,9]. The quantity launched from the food matrices, which can be solubilized in the lumen of the intestine and become available after digestion in the upper tract to be absorbed in the intestine is referred to as bioaccessibility [10]. Certain elements influence the bioaccessibility and bioavailability of phenolic compounds, including chemical structure, molecular size, solubility, degree of polymerization, their concentration in food, their degree of release from food matrices, and their connection with other molecules (e.g., proteins, lipids, and fibers), different food processing methods, as well as digestive enzymes and microbiota [11]. *Vigna,* as one essential legume, has high complex food matrix. They possess more substantial protein contents (20–24%), dietary fibers (13–17%), and minerals (1–2%) than other main grains [2]; they also include various anti-nutrients such as phytate, hemagglutinin, and protease inhibitors [12]. All these compounds may have impacts on *Vigna* bioaccessibility. For example, macronutrients could combine with specific phenolic compounds due to physical trapping structures, thus sustaining the release of such bioactive compounds from the food matrix [10]. Further, trypsin inhibitors and phytates could reduce the digestibility of proteins and mineral absorption, related to the enzyme activity or chelating properties [12].

In vitro digestion is able to simulate the human body, and assess the nutrients along with bioactive components’ bioaccessibility in foods after ingestion [13]. It has been considered that in vitro digestion will promote the bioaccessibility of dietary phenolics with the process to a certain extent [14]. Following bean consumption, the stomach and small intestine would first digest and absorb some dietary polyphenols and metabolites with low efficiency [11]. The remainder of the undigested food pellet would enter the colon, in which the existing unabsorbed nutritional constituents would be digested and used primarily throughout fermentation by microbial organisms. Microbes in the gut may be capable of converting complex phenolic compounds into simple structured compounds with a low molecular weight that are easier to absorb [15,16]. Nevertheless, research about in vitro digestion effects on bioactive compounds’ bioaccessibility in different *Vigna* species is very limited. The information about bioaccessibility alterations of *Vigna* pulses after colonic fermentation could not be found either. Therefore, this study intended to evaluate phenolic compounds’ antioxidant capacity as well as bioaccessibility changes from four different *Vigna* beans (adzuki bean, black urid whole, black eye beans, and mung bean) using in vitro model of gastrointestinal digestion and colonic fermentation. Moreover, the metabolites formed through colonic fermentation of these *Vigna* samples (SCFAs) were also measured. 

## 2. Materials and Methods

### 2.1. Sample Preparation

Four varieties of *Vigna* samples (adzuki bean, black urid whole, black eye beans, and mung bean) were gained from the local market in Melbourne, VIC, Australia. An electric grinder (Sunbeam Multi Grinder, EM0405, Melbourne, VIC, Australia) was then utilized to grind the legumes into fine powder, with an average particle size of 495 μm. All the legume powder samples were preserved in a dark and dry place at room temperature and were ready for further extraction.

### 2.2. Chemicals and Reagents

Analytical grade chemicals were utilized for extraction and characterization of *Vigna* pulses. The 98% sulfuric acid was gained from RCI Labscan Ltd., Bangkok, Thailand. The sodium carbonate anhydrous was acquired from Chem-Supply Pty Ltd., Adelaide, SA, Australia. The majority of analytical grade chemicals, along with standards utilized in this research were obtained from Sigma-Aldrich (Castle Hill, NSW, Australia), containing quercetin, gallic acid, catechin, vanillin, and 6-hydroxy-2,5,7,8-tetramethylchrom an-2-carboxylic acid (Trolox), 2,2′-diphenyl-1-picrylhydrazyl (DPPH), Folin–Ciocalteu reagent, 2,4,6-tripyridyl-s-triazine (TPTZ), hydrated sodium acetate, hexahydrate aluminum chloride, ferric chloride (Fe (III) Cl_3_·6H_2_O), along with potassium chloride, sodium chloride, potassium dihydrogen phosphate, sodium hydrogen carbonate, magnesium dichloride, sodium hydroxide pellets, ammonium carbonate, calcium chloride, hydrogen chloride, pepsin, as well as bile salts. The in vitro digestive enzymes, such as α-amylase and pancreatin, were bought from US Biological (Assay Matrix Pty Ltd., Clyde, VIC, Australia). Milli-Q water utilized in this research was from Millipore Milli-Q Gradient Water Purification System (Darmstadt, Germany).

### 2.3. Extraction of Phenolic Compounds from Raw Materials

Free as well as bound phenolic compounds both were extracted from *Vigna* samples in terms of the protocols discussed by Peng, et al. [17] with slight changes. Three-gram of *Vigna* powder was homogenized thoroughly in water (1:10, *w/w*) using Ultra-Turrax T25 Homogenizer (IKA, Staufen, Germany) persisting thirty seconds at a speed of 10,000 rpm, accompanied with a 12-h incubation using the shaking incubator (ZWYR-240 incubator shaker, Labwit, Ashwood, VIC, Australia) via 4 °C at 120 rpm. The mixture was then centrifuged utilizing Hettich Refrigerated Centrifuge (ROTINA380R, Tuttlingen, BadenWürttemberg, Germany), continuing for fifteen minutes under a speed of 8500 rpm. The supernatant containing free phenolic extracts was collected as free phenolic extract after passing it through a syringe filter with 0.45 µm (PVDF, Millipore, MA, USA). Following that, the solid residue was alkaline and acidic hydrolyzed. The mixture’s supernatant was obtained as bound phenolics extract and filtered under the same circumstances. For further analysis, the entire phenolic extracts were kept at –20 °C.

### 2.4. In Vitro Gastrointestinal Digestion

Based on Gu, et al. [18], the in vitro digestion of four *Vigna* pulses was conducted in the order of oral, gastric, along with intestinal phases utilizing simulated oral (SOF), gastric (SGF), and intestinal (SIF) fluids in accordance with the INFOGEST 2.0 protocol. Pulses powder was added to water with a proportion of 1:2 (*w/v*), and a non-digested aliquot of 5 mL was taken. Following that, SOF (1:1, *v/v*) was transferred to the mixture to alter the pH value to approximately 7.0, accompanied by adding 75 U/mL of salivary α-amylase, also 2 min incubation under 37 °C with continuing shaking. The oral phase aliquot was then taken at 5 mL. The gastric phase was mimicked by combining SGF (1:1, *v/v*) in the oral bolus (1:1, *v/v*) with 2000 U/mL of porcine pepsin. The mixture pH was changed to approximately 3.0 via HCL addition, followed by a 2-h incubation at 37 °C. The aliquot of gastric digestion was 5 mL, which was terminated by raising the pH to 7.0. For initiating intestinal digestion, SIF (1:1, *v/v*), 100 U/mg trypsin, along with 10 mM bile salt were all introduced to the gastric aliquot, accompanied by incubation under identical conditions. To stop the enzymic reactions, each digestion phase aliquots were frozen via liquid nitrogen directly after completing incubation.

### 2.5. In Vitro Colonic Fermentation

In the presence of human gut microbiota, the non-absorbed portion of gastrointestinal (GI) digestion was persistently fermented according to Gu, Suleria, Dunshea and Howell [18] with slight alterations. Pig feces were needed as a substitute for human feces from 10 mixed male and female large landrace grower white pigs (50 kg live weight). They have been fed a two-week standard diet and grown in the Diamond Valley Pork animal house (Thomas Road, Laverton North VIC, Australia). Fresh feces samples were obtained and needed to mix within one anaerobic chamber. Twenty percent of feces consisted of homogenizing twenty grams of feces with eighty-gram 0.1 M sterilized pre-nitrogen flushed phosphate buffer (pH = 7.0) utilizing a stomacher mixer persisting five minutes (MiniMix^®^ Lab Blender, Thomas Scientific, Swedesboro, NJ, USA). The medium was then filtered through a sterile muslin cloth. The pellet of intestinal digesta was made via centrifugation at a rate of 10,000× *g* for ten minutes and combined with 5 mL fecal medium along with basal media in 6 sets of nitrogen-flushed tubes. Then the tubes would be incubated in the dark for 0, 2, 4, 8, 16, and 24 h, shaking at 120 rpm. After centrifuging for ten minutes at 10,000× *g* under 5 °C, the supernatant was available for phytochemical bioactivity as well as SCFAs production analysis.

### 2.6. Estimation of Phenolic Content and Antioxidant Capacity

#### 2.6.1. Determination of Total Phenolic Content (TPC)

The total phenolic content of *Vigna* pulses was evaluated according to Mussatto et al. [19] followed by certain changes. In a 96-well plate, gallic acid standard (0–200 μg/mL) or *Vigna* extract, Folin–Ciocalteu reagent solution, along with water (1:1:8, *v/v/v*) were transferred in sequence, followed by a 5-min incubation under room temperature in the dark. With the same sample volume, sodium carbonate (10%, *w/w*) was transferred, and incubated lasting for one hour in the same situations. A UV-VIS spectrophotometer (Thermo Fisher Scientific, Waltham, MA, USA) was then adopted to read the absorbance at 765 nm. Triple independent analysis data were described as mean mg gallic acid equivalents (GAE) per gram in terms of dry weight (mg GAE/g) ± standard deviation (SD).

#### 2.6.2. Determination of Total Flavonoids Content (TFC)

The total flavonoids content of *Vigna* was estimated based on Suleria et al. [20] by certain alterations. In brief, *Vigna* extract or quercetin standard (0–50 μg/mL), 2% aluminum chloride along with 50 g/L sodium acetate (1:1:1.5, *v/v/v*) were transferred to 96-well plate in sequence. Then 2.5-h of incubation under room temperature in darkness was processed. At 440 nm, the absorbance was read. Triple independent analysis data were stated as mean mg quercetin equivalents (QE) per dry weight (mg QE/g) ± SD.

##### 2.6.3. Determination of Total Condensed Tannins (TCT)

The total condensed tannins content of *Vigna* was estimated based on Suleria, Barrow and Dunshea [20]. In brief, the assay was carried out by transferring *Vigna* extract or catechin standard (0–1 mg/mL), vanillin solution, and 32% sulfuric acid (1:6:1, *v/v/v*) to 96-well plate in sequence. Then a 15-min incubation under room temperature in the dark was conducted. At 500 nm, the absorbance was read. Triple independent analysis data were described as mean mg catechin equivalents (CE) per dry weight (mg CE/g) ± SD.

##### 2.6.4. 2,2′-Diphenyl-1-Picrylhydrazyl (DPPH) Antioxidant Assay

The free radical scavenging activity of *Vigna* varieties was measured based on Suleria, Barrow and Dunshea [20] study. DPPH and methanol (1:25, *w/v*) were combined to prepare the 0.1 mM DPPH radical solution. The assay was carried out by pouring 40 μL of *Vigna* extract or Trolox standard (0–200 g/mL) and 260 μL of DPPH solution to a 96-well plate in sequence. Then an incubation for 30 min under 25 °C was performed. At 517 nm, the absorbance was read. Triple independent analysis data were stated as mean mg Trolox equivalents (TE) per dry weight (mg TE/g) ± SD.

##### 2.6.5. Ferric Reducing Antioxidant Power (FRAP) Assay

The FRAP evaluation of *Vigna* was measured according to Suleria, Barrow and Dunshea [20]. In the dark, 300 mM sodium acetate, 10 mM TPTZ solution, along with 20 mM Fe (III) solution (10:1:1, *v/v/v*) were combined to make the FRAP dye solution. In a 96-well plate, 20 μL *Vigna* extract or Trolox standard (0–200 μg/mL) and 280 μL dye solution were transferred. Then an incubation for 10 min under 37 °C was conducted. At 593 nm, the absorbance was read. Triple independent analysis data were stated as mean mg Trolox equivalents (TE) per dry weight (mg TE/g) ± SD.

### 2.7. Quantification of Phenolic Compounds through High-Performance Liquid Chromatography Photodiode Array (HPLC-PDA)

The concentrations of the targeted phenolic compounds in *Vigna* were determined using an Agilent 1200 series HPLC (Agilent Technologies, CA, USA) provided with a photodiode array (PDA) detector along with our previously published methods of Suleria, Barrow and Dunshea [20] and Wu et al. [21]. A Synergi Hydro-RP (250 × 4.6 mm i.d.) reversed-phase column with 4 µm (Phenomenex, Lane Cove, NSW, Australia) was protected by a Phenomenex 4.0 × 2.0 mm i.d., C18 ODS guard column. At the PDA detector, wavelengths of 280, 320, and 370 nm were chosen in parallel. In terms of instrument control, statistics collection, and chromatographic preparation, Empower Software (2010) was utilized.

### 2.8. Gastrointestinal Digestion and Colonic Fermentation Indices

#### 2.8.1. Bioaccessibility of Phenolic Compounds

Across in vitro digestion along with colonic fermentation, each phenolic compound bioaccessibility was predicted as the percentage of compound released from the sample in every digestion stage and compound in the undigested one. Bioaccessibility was calculated using the equation below:Oral Bioaccessibility (%) = (Oral fraction/Total phenolic content) × 100(1)
Gastric Bioaccessibility (%) = (Gastric fraction/Total phenolic content) × 100(2)
Intestinal Bioaccessibility (%) = (Intestinal fraction/Total phenolic content) × 100(3)
Colonic Bioaccessibility (%) = (Colonic fraction/Total phenolic content) × 100(4)

#### 2.8.2. Recovery Index

The recovery index was determined utilizing the following equations applying the soluble and insoluble phenolic fractions of the entire *Vigna* digesta:Soluble (%) = (Content in soluble fraction/Total phenolic content) × 100(5)
Insoluble (%) = (Content in insoluble fraction/Total phenolic content) × 100(6)
Recovery Index (%) = Soluble (%) + Insoluble (%)(7)

#### 2.8.3. Residual Intestinal Digesta Index

The residual phenolic compounds of intestinal digest (RID %) is the unbioaccessible fraction that maintains intact after completing the intestinal phase, which was determined using the equation below:RID (%) = (Intestinal insoluble fraction/Total phenolic content) × 100(8)

#### 2.8.4. Residual Colonic Digesta Index

The residual phenolic compounds of colonic digesta (RCD %) is the unbioaccessible fraction after completing colonic fermentation, and was determined utilizing the below equation:RID (%) = (Colonic insoluble fraction/Total phenolic content) × 100(9)

### 2.9. Short Chain Fatty Acids (SCFAs) Analysis

The SCFAs analysis was carried out in accordance with our previously published protocol according to Gu, Suleria, Dunshea and Howell [18] and Wu, Liu, Lu, Barrow, Dunshea and Suleria [21]. Analytical standard curves were created using acetic, propionic, butyric, iso-butyric, and valeric acids. The SCFAs analysis was performed with the application of gas chromatography (7890B Agilent, Santa Clara, CA, USA) provided with a flame ionization detector (GC-FID), an autosampler (Gilson GX-271, Gilson Inc., Middleton, WI, USA) and also autoinjector. The capillary column (SGE BP21, 12 × 0.53 nm internal diameter with 0.5 µm film thickness, SGE International, Ringwood, VIC, Australia, P/N 054473) with retention gap kit (including a 2 × 0.53 mm ID guard column, P/N SGE RGK2) was adopted for evaluating the SCFAs concentrations. The injection volume applied was 1 µL. The carrier gas was helium, at a total flow speed of 14.4 mL/min, and the mixed gas was composed of nitrogen, hydrogen, and air, at the rates of 20, 30, and 300 mL/min, separately. The temperature for the oven remained at 100 °C persisting thirty seconds, then enhanced to 180 °C for one minute at a speed of 6 °C/min and ultimately kept at 200 °C for ten minutes with a rate of 20 °C/min. The injection port, along with detector temperatures were set to 200 and 240 °C, separately. For statistical analysis, all data were described as mmol/L.

### 2.10. Statistical Analysis

One-way ANOVA and Tukey’s test were applied to validate the significant differences in phenolic content along with antioxidant capacity among distinct varieties and digestion phases (*p* ≤ 0.05). Minitab 19 (Minitab^®^ for Windows Release 19, Minitab Inc., State College, AR, USA) together with GraphPad Prism 9 (Prism 9.0.0, GraphPad Software Inc., San Diego, CA, USA) were adopted for the data analysis and tests. All shown results were subtracted either control or blanking values and stated as mean ± standard deviations (SD) of triple independent analysis.

## 3. Results and Discussion

### 3.1. Phenolic and Bioactivity Changes during In Vitro Digestion

#### 3.1.1. Phenolic Estimation

The total phenolic, flavonoid, and tannin content of four in vitro digested beans are measured as exhibited in Figure 1a–c. Based on the data, the total phenolics content of most tested *Vigna* samples, as well as flavonoids in most *Vigna* (excluding AB), showed a generally increased tendency along with the in vitro digestion. Among the tested samples, adzuki beans (AB) exhibited a relatively higher total phenolic content (TPC) value in the simulated intestine phase (4.76 mg GAE/g). Since finishing the intestinal digestion, the total flavonoid content (TFC), as well as total condensed tannin (TCT), of BEB were relatively higher, showing 0.74 mg QE/g and 10.43 mg CE/g, respectively. 

There was no significant variation in TPC values of AB and BUW from the oral to gastric phase. In Chait, et al. [22] study of simulated gastrointestinal digestion of carob polyphenols, a similarly marginal alteration via the oral phase was detected compared to the undigested one. Just 12% of decrease in TPC along with 25% of TFC was detected, which might be due to the less contact time during the oral phase (2 min) as well as the minimal influence (1.25 μkat/mL) of α-amylase that initiates starch hydrolysis in the mouth [22,23]. Differently, as for MB, the gastric digestion significantly improved the TPC, which is consistent with the observations of Cárdenas-Castro, et al. [24], in which, when compared to the undigested material, the gastric digestion of two common beans led to a 10-fold rise in the determined phenolic content. This may be due to the small pH within the gastric digestion increasing the richness of polyphenols in undissociated form, thereby driving the release from the matrices to the aqueous phase [25]. Moreover, the four studied *Vigna* samples all exhibited a marked increase in TPC after intestinal digestion. Lafarga, et al. [26] also indicated the in vitro digestion resulted in increased TPC of all studied eight pulse species (i.e., lentil, cowpea, faba bean, chickpeas, soybean, etc.). During the intestinal digestion stage, the connections between carbohydrates and phenolic compounds could be lessened below neutral pH environments (6.9) and the appearance of α-amylase, pancreatin, lipase, along with bile salts, enabling the transfer and bioaccessibility of phenolic compounds [27]. Furthermore, microflora existing in the digestive system also plays an important role [24].

As for total flavonoid content (TFC), similarly, an incredible increase during the stage of intestinal digestion was detected for all *Vigna* samples, which is related to Zhang et al. [28] findings. However, all tested *Vigna* samples underwent a notable decrease after oral digestion, possibly attributed to the particular exposure time in the oral phase and the minimal influence of α-amylase as explained above. Based on past findings, the hydrolysis of glycoside flavonoids begins in the mouth due to the action of β-glycosidase; nonetheless, its performance is sensitive to the kind of sugars available in the molecule [29]. In contrast to others, such as rhamnose conjugates, glucose conjugates are speedily hydrolyzed [30]. Flavonoids, as phenolic compounds, constitute both non-covalent as well as covalent connections with proteins in human saliva, determined by the size of the phenolics [31]. Based on the literature, black eye beans (BEB), mung bean (MB), and adzuki beans (AB) are all abundant in rutin, myricetin, and quercetin [3]. Therefore, it can be presumed that after oral digestion, the reduction of these predominant flavonoid compounds directly caused a TFC decrease in our tested *Vigna* digesta. In addition, it is worth noting that the TFC values in gastric digestion were not detected in mung beans, black urid whole (BUW), and black eye beans. This might be because the flavonoids interact with protease under a small pH value and form flavonoid-protease complexes [32]. Simultaneously, the flavonoid oligomers will be degraded to smaller units during the gastric phase [29]. 

The values of total condensed tannins content (TCT) in the undigested phase were traced or even not detected. The hydrolysable tannins as well as high-molecular-weight proanthocyanidins account for over 75% of all food polyphenolic compounds consumed on average, and they may attach closely to dietary fibers, limiting their accessibility [33]. According to Luo et al. [34], the TCT value of MB was detected as 11.23 mg CE/g, and that in AB was 19.17 mg CE/g, much higher than our findings. While similar to our result, no tannin was found in MB in Price et al. [35] experiment. The different TCT results in different studies may be owing to the differences in terms of tannin extraction solvents and methods, and sample processing as well. Secondly, the TCT values in four *Vigna* samples were significantly promoted during oral phase digestion, while that in the gastric phase of all *Vigna* samples were not detected. This may be attributed to the acidic condition of the gastric phase having adverse effects on condensed tannins in *Vigna*; moreover, condensed tannins might be precipitated by pepsins through hydrogen bonding and hydrophobic interactions. More importantly, after intestinal digestion, the TCT values of AB and BEB were enhanced significantly, suggesting the probable positive effects of intestinal digestion on condensed tannins. However, in general, the food components that humans ingest are in complex matrices, while in our study, the phenolic compounds were evaluated in isolation, which could be a certain limitation in our research.

#### 3.1.2. Antioxidant Activities Estimation

In the presented study, the DPPH and FRAP assays were accessed to evaluate the antioxidant activity during the different in vitro phases. The DPPH assay is based on an electron transfer and measures the inhibition of nitrogen free radicals, while the FRAP assay is different, as there are no free radicals involved. This assay is based on monitoring the reduction of ferric iron (Fe^3+^) to ferrous iron (Fe^2+^) [36]. The antioxidant activities of in vitro digested *Vigna* pulses are exhibited in Figure 1d,e. AB demonstrated relatively higher free radical scavenging capacity (DPPH) as well as ferric reducing antioxidant power (FRAP) during the in vitro digestion. Moreover, it is noticeable that except for the intestinal DPPH value not detected in BUW and BEB samples, along with remarkable increase in FRAP value of AB after intestinal digestion, almost all samples showed generally reduced DPPH and FRAP values after in vitro digestion. 

The highest antioxidant activity was found in the undigested phase and the lowest was observed in the oral and gastric phases for all samples. During the intestinal phase, the AB sample showed the highest antioxidant activity both by DPPH (~3.5 mg TE/g) and by FRAP (~2.5 mg TE/g). In the intestinal phase, MB showed antioxidant activity by the two methods tested, while BUW and BEB only detected the presence of phenolic compounds by the FRAP method. This difference in the detection of antioxidant activity can be attributed to the difference in the way of measuring each of the antioxidant analyses. The antioxidant activity estimation results suggested that after the oral phase, the antioxidant activity could be more related to flavonoids and to a lesser extent to phenolic acids in the samples. Based on the results of TFC as mentioned in the previous part (3.1.1. Phenolic Estimation), the content of flavonoids was remarkably decreased during oral and gastric phases, which may be the dominant reason for significantly decreased DPPH and FRAP values in our study. In addition to the hydrolysis and covalent effects of flavonoids, the digestibility along with absorption of other certain polyphenols, could be significantly influenced by the mucins, salivary albumin, and proline-abundant proteins during oral digestion [29]. Furthermore, during the gastric digestion, the most severely impacted phenolic class, including flavan-3-ols, accompanied by ellagitannins, gallotannins along with anthocyanins, was detected in *Arbutus unedo* [23].

Likewise, in Chait, Gunenc, Bendali and Hosseinian [22] experiment of the carob in vitro digestion, the antioxidant capacity of phenolics with bound, soluble-free, soluble-conjugated forms was analyzed. The DPPH, ABTS along with ORAC values of soluble free phenolic compounds all showed a marked increase after intestinal digestion, when compared to undigested samples and oral and gastric digestion. Further, the antioxidant capacity (in terms of DPPH, ABTS along with ORAC values) of soluble conjugated and bound phenolic compounds exhibited gradual and significant reduction during the in vitro digestion [22].

In addition, most tested samples did exhibit significant increase in DPPH and FRAP after intestinal digestion. This suggests that intestinal digestion promotes antioxidant capacity in tested *Vigna* samples and is consistent with the improving effect of intestinal digestion on the TPC, TFC and TCT values as mentioned above. The result is also compatible with the Zhang, Deng, Tang, Chen, Liu, Dan Ramdath, Liu, Hernandez and Tsao [28] findings, in which the antioxidant capacities in terms of DPPH as well as FRAP of lentils largely improved to 16.40 µmol TE/g DW along with 15.00 µmol AAE/g DW, separately after completing the intestinal phase, while, noticeably, these data are found significantly less than the undigested extracts [28]. Because the gastric and intestinal tracts are constantly subjected to intense oxidative stress caused by free radicals as well as reactive oxygen species developed from microbes and toxins in the diet, antioxidants such as phenolics issued from the food material into the digestive tract might indeed play a key part in reducing oxidative stress [37].

### 3.2. Phenolic and Bioactivity Changes during Colonic Fermentation

#### 3.2.1. Phenolic Estimation

The alterations of total phenolic compounds, total flavonoids and condensed tannins content in *Vigna* digesta through colonic fermentation are demonstrated in Figure 2a–c. After 2 h of reaction, all TPC values significantly increased and attained the highest in four *Vigna* digesta, implying the positive effect of phenolic releasement by colonic fermentation. According to Acosta-Estrada, et al. [38], undigested polyphenolic compounds, especially those covalently attached to food matrices, are resisted by digestive enzymes along with the approach to the colon, in which they can be catabolized via existing microflora and degraded to phenolic acids because of enzymatic releasement [29]. The growing effect upon TPC at the initial colonic stage in our study is consistent with Chait, Gunenc, Bendali and Hosseinian [22] results, in which a general rise across the first 5 h of fecal reaction. Subsequently, TPC values significantly reduced after 4-h fermentation, followed by specific rise after 8-h fermentation, and gradually diminished after 16-h and 24-h fermentation. This may be due to the colonic microbes’ further decomposition and absorption of the relevant phenolic compounds. The general diminished effect during the later stages within colonic fermentation on total phenolics was consistent with Barros, et al. [39] experiment. The TPC values of two studied cowpea cultivars were observed significantly decrease after colonic fermentation. In Alqurashi et al. [40] study, after 24 h of colonic fermentation, a reduction or complete degradation of phenolic acids in acai appeared.

The total flavonoid content (TFC) during colonic fermentation showed a generally increasing trend from initial until 8 h in all tested *Vigna* digesta. It is also noticed that the TFC of AB reached the highest after 4-h fermentation, and for MB it reached the upper point after 2-h, while after 8 h until 24 h, the TFC in MB showed a gradually decreasing trend, and TFC in the other three pulses all dropped to zero. The metabolism and transformation of certain flavonoid compounds may cause an overall increase in TPC during early colonic fermentation. According to the literature, just 2–15% of dietary flavonoids are digested in gastrointestinal digestion [41]. Bacterial glycosidases, glucuronidases, as well as sulfatases in the colon were able to eliminate all rest glycosides, glucuronides, along with sulphates from the flavonoid moiety. The flavonoid aglycon can be deeper metabolized by bacteria to produce ring fission products containing valerolactones and a variety of low-molecular-weight phenolic acids [42].

As for total condensed tannins (TCT) in *Vigna* digesta, they generally showed declined trends in tested four *Vigna* digesta. The TCT values of AB and BUW decreased to zero after 4 h of fermentation. The TCT value of MB also showed a marked decrease during the whole fermentation stage. Regarding Saura-Calixto, et al. [43], 46% of dietary tannins become bioaccessible in the large intestine. Previous studies also confirmed that proanthocyanidins are greatly metabolized by gut microflora and are the most bioaccessible tannins in the large intestine [43,44]. Polymeric proanthocyanidins could not be analyzed after 48-h of colonic fermentation as reported [15]. Further, interactions between tannin metabolites, other constituents existing in the fermented residue, along with assay chemical products induce precipitation and, ultimately, below the limit values of TCT evaluation. Generally, polyphenols increase the population of beneficial bacteria such as *Bifidobacterium* and *Lactobacillus*, which help to protect the intestinal barrier [44]. Polyphenols have been shown to have prebiotic properties as well as antimicrobial properties against pathogenic gut microflora. Evidence suggests that dietary polyphenols have shown advantages in a variety of disorders, as well as a significant impact on the gut microbiota in the direction of symbiosis [11,40].

#### 3.2.2. Antioxidant Capacity Estimation

The alterations in the antioxidant potential of *Vigna* through colonic fermentation are predicted by DPPH along with FRAP assays, which are displayed in Figure 2d,e. A similar fluctuation in DPPH was discovered across all four legumes, which significantly expanded after 2 h of fecal reaction and then abruptly decreased after four hours of reaction. The DPPH values grew substantially and remained relatively constant until 24 h of reaction. It was discovered that the free radical scavenging capacity of *Vigna* pulses could be associated with the presence of phenolics, particularly flavonoids. Similarly, in Goderska et al. [45] study, the antioxidant capacities in Jaś Karłowy bean flour and its extruded products were elevated through colonic fermentation with specific microflora, and significantly higher than in intestinal phase and the undigested.

On the other hand, as for FRAP assay, although the values for 8-h colonic fermentation in AB, 2-h colonic fermentation in MB, 4-h colonic fermentation in BUW had reached their highest points, respectively, there was an overall decreasing trend. Similarly, Barros, Abreu, Rocha, Araújo and Moreira-Araújo [39] reported the general decrease effect of colonic fermentation on both FRAP and ABTS values in two studied cowpea cultivars. Moreover, in Dong, Liu, Xie, Chen, Zheng, Zhang, Zhao, Wang, Xu and Yu [14] study, the antioxidant capacities were reduced slightly at 24 h of fermentation; these findings suggested that free polyphenols during the initial fermentation could be degraded and catabolized until being digested by colonic microbes after 12 h of fecal reaction.

### 3.3. Bioaccessibility of Individual Phenolic Compounds in Four Vigna Samples

The bioaccessibility of seven individual phenolic compounds across various in vitro digestion phases of the tested *Vigna* samples is illustrated in Table 1. The portion of consumed bioactive components which can be digested by the gastrointestinal epithelial layer is termed bioaccessibility [21]. According to the data results, the digestion phase observed for highest or lowest bioaccessibility value of phenolic compounds in different *Vigna* samples is different. However, the total phenolic compounds’ bioaccessibility were found to be relatively higher for most *Vigna* samples (AB, MB, BUW) during oral phase digestion.

As for AB, gallic acid, chlorogenic acid along with *p*-hydroxybenzoic acid obtained the highest bioaccessibility in the oral phase, along with pretty high bioaccessibility value of total phenolic compounds. However, the lowest bioaccessibility of its total phenolics was detected during the intestinal phase. This is because of the considerably low accessibility of gallic acid and *p*-hydroxybenzoic acid, which both turned into the lowest values since gastric digestion. 

Several studies confirmed the promoted bioaccessibility of gallic acid by oral digestion, followed by certain diminished effects by the gastric phase. Lucas-González et al. [46] noted the behavior of gallic acid in tested persimmon flour added to spaghetti; shortly, a significant growth was recorded in the mouth phase, accompanied by a decline in the gastric as well as intestinal phases. For post-oral digestion, the greater gallic acid amount of *Jomara dates* was discovered, indicating a more than 100 percent secretion compared to the previous gallic acid amount of dried dates [47]. Agudelo et al. [48] also reported a much higher gallic acid bioaccessibility in the mouth phase compared to the gastric phase in Andean berries. 

Past research also noted that the bioaccessibility of *p*-hydroxybenzoic acid and chlorogenic acid change after in vitro digestion. As the dominant phenolics, chlorogenic acid displayed the highest bioaccessibility after oral digestion of coffee beans [21]. In Zhu et al. [49] experiment of three legumes during in vitro digestion, the bioaccessibility of chlorogenic acid also decreased after the gastric digestion compared to that in the initial oral phase. However, *p*-hydroxybenzoic acid of the studied soybean, kidney bean and faba bean showed increased trends after the gastric digestion in their study, which due to the phenolic compounds launched across the in vitro digestion stages were different in various legumes.

Moreover, as for the lowest bioaccessibility of total phenolics in AB in the intestinal phase, which attributed that the polyphenols are extremely delicate to the slightly alkaline environment encountered in the small intestine, where the majority of polyphenolic compounds are broken down or transferred into some other compounds [29]. Furthermore, digestive enzymes in the intestine contribute to the secretion of phenolics from the food matrices, which appears to be detrimental to the structural rigidity of the polyphenolic compounds and thus subjected to hydrolysis [50]. Figure 3 depicts a graph of the metabolic routes of certain phenolic compounds in legumes.

The least bioaccessibility value of phenolics in MB was found in gastric phase digestion, owing to the zero bioaccessibility of the caftaric acid, *p*-hydroxybenzoic acid and coumaric acid obtained at the gastric step. In Chait, Gunenc, Bendali and Hosseinian [22], although the accessibility of soluble free caffeic acid in gastric phase digestion showed slight increase, its soluble-conjugated and insoluble-bound forms both decreased significantly, which may explain the lowest gastric bioaccessibility of caftaric acid for MB in our finding. According to Panagopoulou, Chiou, Kasimatis, Bismpikis, Mouraka and Karathanos [47], the caffeic acid after gastric digestion became almost undetectable in the *Safawi* date sample, indicating the negative effect of the gastric phase on caffeic acid. Conversely, Lafay and Gil-Izquierdo [51] demonstrated that the caftaric acid is mainly absorbed from the stomach, which may result from the different food matrices in our study. Further, for *p*-coumaric acid bioaccessibility in reported carob, its soluble free and bound forms both significantly reduced to the lowest in gastric phase, along with its soluble conjugated form reduced, which is consistent with the lowest gastric bioaccessibility of MB coumaric acid in our result [22]. Moreover, the minimum bioaccessibility of phenolics in BUW was illustrated after completing the fermentation in the colon, although the protocatechuic acid bioaccessibility was obtained the highest at the colonic stage. Consistently, Chait, Gunenc, Bendali and Hosseinian [22] indicated that during colonic fermentation, the bioaccessibility of kaempferol in carob decreased to non-detected.

As for BEB, the highest bioaccessibility value of its total phenolics was achieved at the gastric phase, mainly owing to the releasement of kaempferol during the gastric digestion stage. The strong stability during the gastric phases of flavonols or flavan-3-ols during simulated in vitro and in vivo digestion are reported, such as the study of chokeberry juice [52]. It was also mentioned that an acid pH environment in the gastric digestion provides protection for polyphenols from breakdown [53].

Lastly, according to Cárdenas-Castro, Pérez-Jiménez, Bello-Pérez, Tovar and Sáyago-Ayerdi [24], chlorogenic acid was initially recognized as the primary polyphenol launched during in vitro digestion, accompanied by kaempferol 3-*O*-glucoside along with quercetin 3-*O*-glucoside, comparable to our study, in which the chlorogenic acid and kaempferol showed the highest bioaccessibility during oral and gastric phase, separately. Specifically, our study demonstrates that the bioaccessibility of chlorogenic acid in all *Vigna* samples was observed to be the highest in salivary digestion. The bioaccessibility of kaempferol of BEB raised from 0% to 102.60% in gastric phase, indicating its sufficient releasement by gastric digestion. Furthermore, the differences between bioaccessibility (%) variations of individual and total phenolic compounds in each *Vigna* through the entire in vitro digestion along with colonic fermentation, may result from the content differences in terms of original phenolic compounds along with other non-nutrient and nutrient compositions among distinct *Vigna* samples.

### 3.4. Intestinal Recovery and Residual Index of Individual Phenolic Compounds in Four Vigna Samples

The recovery as well as residual index of specific phenolic compounds discovered in *Vigna* samples after intestinal as well as colonic fermentation are exhibited in Table 2. Individual phenolic compound recovery in the intestine and colon was significantly different for most *Vigna* samples (MB, BUW and BEB). Specifically, the recovery index of these three *Vigna*s’ colonic fermentation is noticeably lower than that of intestinal digestion. The colonic fermented residue contained lower concentrations of the individual fraction of the most undigested phenolics that stayed in intestinal digesta, along with considerably lower levels of residual total phenolic compounds in colonic digesta. Both tendencies suggested that different enzymes and microorganisms may convert native phenolic chemicals into minor phenolic metabolites to varying degrees [21]. 

Gallic acid in all *Vigna* samples, together with protocatechuic acid, chlorogenic acid, and kaempferol, were all recovered at 100% during the small intestinal digestion, implying that neither of these molecules was broken down or that some of these molecules were constituted via the breakdown of other complicated parts [21]. At the intestinal level, the bonds among the food matrices’ supramolecular structure and polyphenols (low molecular weight) can be disrupted by pH, enzymes or bile salts, etc., resulting in releasement from the food matrices [54]. According to Ordoñez-Díaz, et al. [55], TPC in mango pulp grows after in vitro intestinal digestion, from 6.17 to 7.25 mmol/g dry weight, with an average bioaccessibility indicator of 207.4 percent. This rise was primarily according to a significant improvement in specific phenolics content after completing the intestinal phase, containing gallic acid, 3-*O*-methylgallic acid, and methyl gallate, as well as the presence of 3,4-dihydroxybenzoic acid, presumably from the monogallyl glucoside as well as hydroxybenzoic acid hexoside hydrolysis. In addition, Mosele, Macià, Romero and Motilva [23] demonstrated the negative value of galloyl quinic along with galloyl shikimic acids, and catechin detected after gastric phase transformed positive after completing the intestinal phase, suggesting neutral pH in the intestinal phase disrupted the connection between phenolic and pectin-gel produced structure according to depolymerization via β-elimination [56]. Moreover, intestinal digestion could elevate the releasement of gallic acid in tested Arbutus unedo [23]. 

Adversely, some studies demonstrated that various stages of in vitro digestion decreased the polyphenols’ recovery, which was different from our finding. For example, a reduced effect on phenolic recovery index within carrot by 35.3% after intestinal digestion was reported [14]. Similar phenomenon was also indicated in Goulas and Hadjisolomou [57] study of carob. The possible reason for the promoting effects on recovery index by intestinal phase in our findings may be because of the existence of dietary soluble fiber in *Vigna* samples, which may remain the stability of phenolics within intestinal stage, thereby preventing their recovery index.

Our study also demonstrates that the colonic fermentation may deeper enhance the launch of phenolic acids from the indigestible *Vigna* portion. For instance, the intestinal recovery indices of *p*-hydroxybenzoic acid were 0 in 4 *Vigna* samples, implying a breakdown of launched *p*-hydroxybenzoic acid present across the intestinal phase. Nonetheless, its colonic recovery index of BEB improved to more than 100%, therefore determining the further release of phenolics via colonic microbes [58]. The appearance of dietary soluble fibres in *Vigna* still maintained the stability of phenolics, thereby protecting their recovery levels [59]. Simultaneously, the recovery index of protocatechuic acid after colonic stage even raised to 204.02%. Correa-Betanzo et al. [60] exhibited that significant levels of phenolic acids including syringic acid, caffeic acid, and kaempferol-3-rhamnoside were generated after completing the fermentation in the colon for wild blueberry. Whereas, the decreased recovery of chlorogenic acid in colon for BEB was detected, which is consistent with Correa-Betanzo, Allen-Vercoe, McDonald, Schroeter, Corredig and Paliyath [60] results.

### 3.5. Short Chain Fatty Acids Production

Five major short chain fatty acids (SCFAs) (acetic, valeric, butyric, propionic, and iso-butyric acid) in four colonic-fermented *Vigna* varieties are determined and displayed in Figure 4. In general, the formation of SCFAs gradually rose after 4-h of reaction and peaked after approximately 16-h fermentation and remained or began decreasing gradually. While as for propionic acid in MB and BUW digesta, it reached the top point after 8-h colonic fermentation (around 0.006 mmol/L), different from that in AB and BEB. The total SCFAs concentration in AB was the highest (0.021 mmol/L), followed by BUW, BEB, and MB. Significant differences in SCFAs production could be noted with distinct *Vigna* species.

On the other hand, it can be generally concluded that the concentration level of generated SCFAs in tested four *Vigna* digesta during the fermentation stage is: acetic acid > iso-butyric acid > propinonic acid. Similarly, in Chen, Chang, Zhang, Hsu and Nannapaneni’s [16] experiment, the acetic acid was detected as the most rich SCFA in tested pinto bean and soybean during the whole fermentation stage. The extent of acetic acid grew dramatically during the first 12 h and after 12 h, acetic acid was produced at a slower speed and tended to maintain a stable level. This general tendency is slightly similar to our findings, rapidly rising within 16 h but remaining after 16 h of fermentation. In the Çalışkantürk Karataş, et al.’s [61] research, after 24 h of reaction, the total percentages of propionic along with butyric acids achieved were marginally greater than 48 percent in two tested faba bean samples, which was much higher than our results (30% and 6%, respectively). Furthermore, acetic was the top producer of SCFAs from all tested substrates, followed by butyric, propionic, and valeric acids. The slight differences in SCFA production may be due to the different legume samples used as well as fermentation conditions.

Pulses contain dietary fibers and oligosaccharides, as well as polysaccharides which are not able to be absorbed by the human digestive tract and may be potential substrates for colonic microflora, such as cellulose, hemicellulose, pectin, gums, along with enzyme resistant starch [62]. The formation of SCFAs can be dominantly correlated to the carbohydrate’s catabolism, mainly the resistant starch along with dietary fibers. Wong et al. [63] mentioned that the substrate origin and gut transit time might affect the rate and amount of SCFAs produced as well. Among the four *Vigna* varieties, the starch content (main polysaccharide in pulses) of AB may be the highest, while that of MB may be the lowest.

## 4. Conclusions

In conclusion, the release of phenolic compounds and their bioaccessibility in four tested *Vigna* samples could be generally enhanced through in vitro digestion and colonic fermentation to a certain extent. Different in vitro digestion along with colonic fermentation stages, exhibited different effects on phenolic content, antioxidant capacities, bioaccessibility, and recovery index for different beans. Among the tested samples, adzuki bean implied comparably higher TPC, FRAP and DPPH values during the in vitro digestion. The behavior of gut microflora launched more phenolics from legume digesta, owing to the breakdown of the trapping structure formed by specific macromolecules. The greatest TPC of all *Vigna* samples was discovered at 2-h colonic fermentation. In addition, the bioaccessibility of total phenolics was comparably higher in most *Vigna* samples (including adzuki bean, mung bean, and black urid whole) during oral phase digestion. Most *Vignas*’ colonic fermentation recovery indices are remarkably smaller than that of intestinal digestion. Moreover, the concentration of acetic acid during SCFA production was the highest, followed by iso-butyric acid and propinonic acid. Adzuki beans may be more beneficial to gut health and possess a relatively higher antioxidant potential after consumption. However, in vitro digestion and fermentation models present some limitations related to the digestion and absorption of nutrients, and the use of different reagents, enzymes and inoculum that can lead to distinct results. For this reason, the application of standardized and reliable in vitro models is necessary. Moreover, due to other compounds existing in *Vigna*, more accurate and detailed information is needed to be further searched and provided, such as the content of individual phenolic compounds as well as other non-nutrient and nutrient compositions, to further conclude the influence of different food matrices on phenolics during digestion. Meanwhile, the biotransformation of phenolic compounds after 48 h of colonic fermentation still needs further exploration.

## Figures and Tables

**Figure 1 foods-11-03884-f001:**
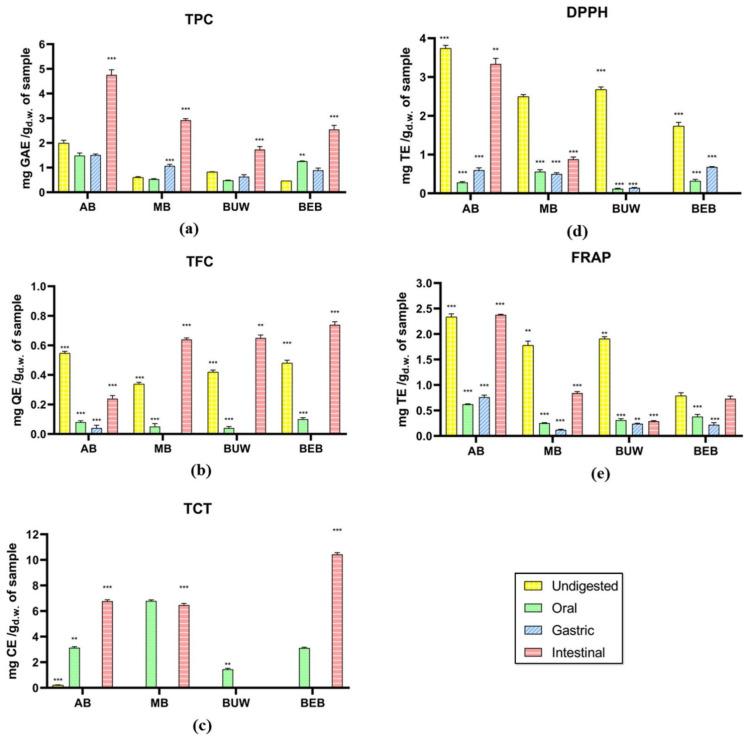
The evaluation of phenolic content and antioxidant capacity of different legumes in vitro digestates. (**a**) Total phenolic content; (**b**) total flavonoids content; (**c**) total condensed tannins; (**d**) 2,2′-diphenyl-1-picrylhydrazyl antioxidant assay; (**e**) ferric reducing antioxidant power assay. The results of all assays were pure values that have been subtracted the values of control. Legume samples mentioned in abbreviations are Adzuki Beans “AB”; Mung Bean “MB”, Black Urid Whole “BUW”, Black Eye Beans “BEB”. Yellow bars: Undigested phase; Green bars: Oral phase; Blue bars: Gastric phase; Pink bars: Intestinal phase. GAE: gallic acid equivalents; QE: quercetin equivalents; CE: catechin equivalents; TE: Trolox equivalents. **: Statistically significant different (*p* ≤ 0.05); ***: Statistically very significant different (*p* ≤ 0.01).

**Figure 2 foods-11-03884-f002:**
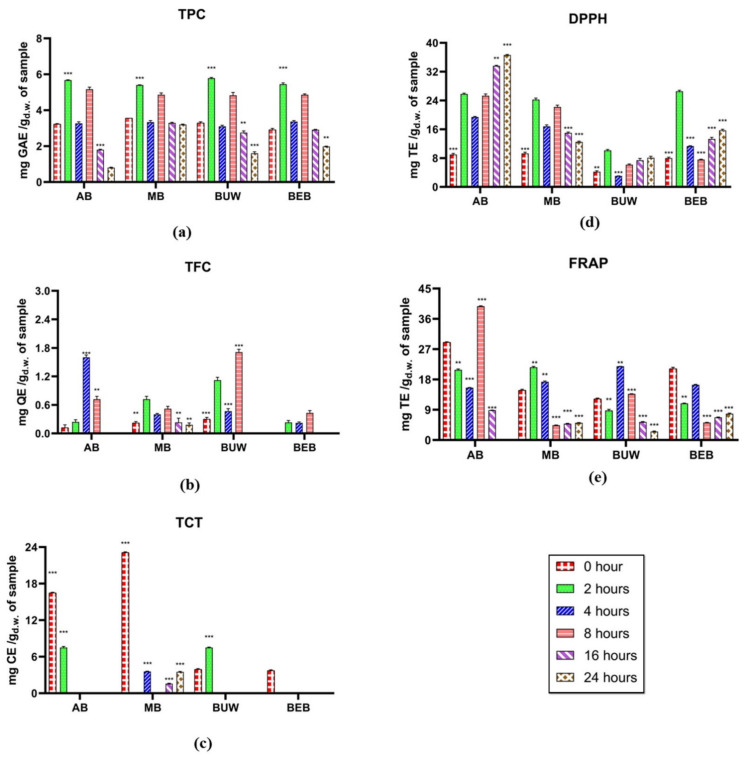
The antioxidant potential estimation of different legumes digesta after complete colonic fermentation. (**a**) Total phenolic content; (**b**) total flavonoids content; (**c**) total condensed tannins; (**d**) 2,2′-diphenyl-1-picrylhydrazyl antioxidant assay; (**e**) ferric reducing antioxidant power assay. The results of all assays were pure values which have been removed the values of control. Legume samples mentioned in abbreviations are Adzuki Beans “AB”; Mung Bean “MB”, Black Urid Whole “BUW”, Black Eye Beans “BEB”. Red bar: 0 h; Green bar: 2 h; Blue bar: 4 h; Pink bar: 8 h; Purple bar: 16 h. Brown bar: 24 h. GAE: gallic acid equivalents; QE: quercetin equivalents. TE: Trolox equivalents. **: Statistically significant different (*p* ≤ 0.05); ***: Statistically very significant different (*p* ≤ 0.01).

**Figure 3 foods-11-03884-f003:**
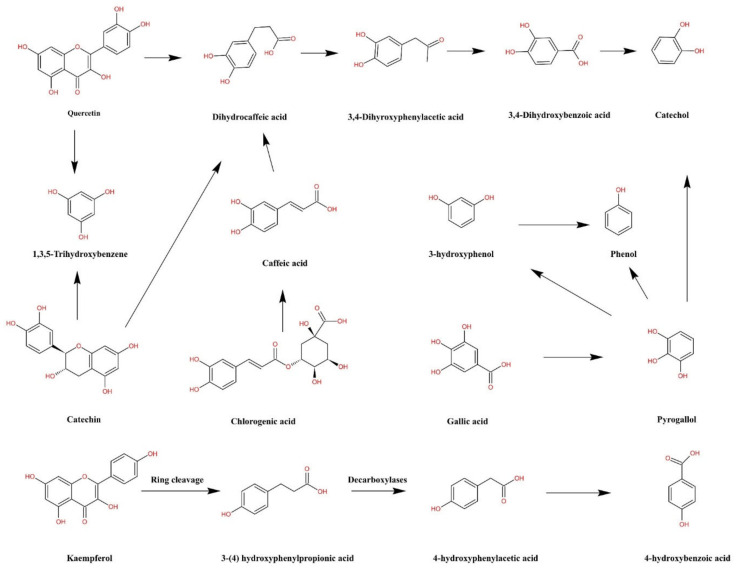
Possible digestion routes of the phenolic compounds in legumes.

**Figure 4 foods-11-03884-f004:**
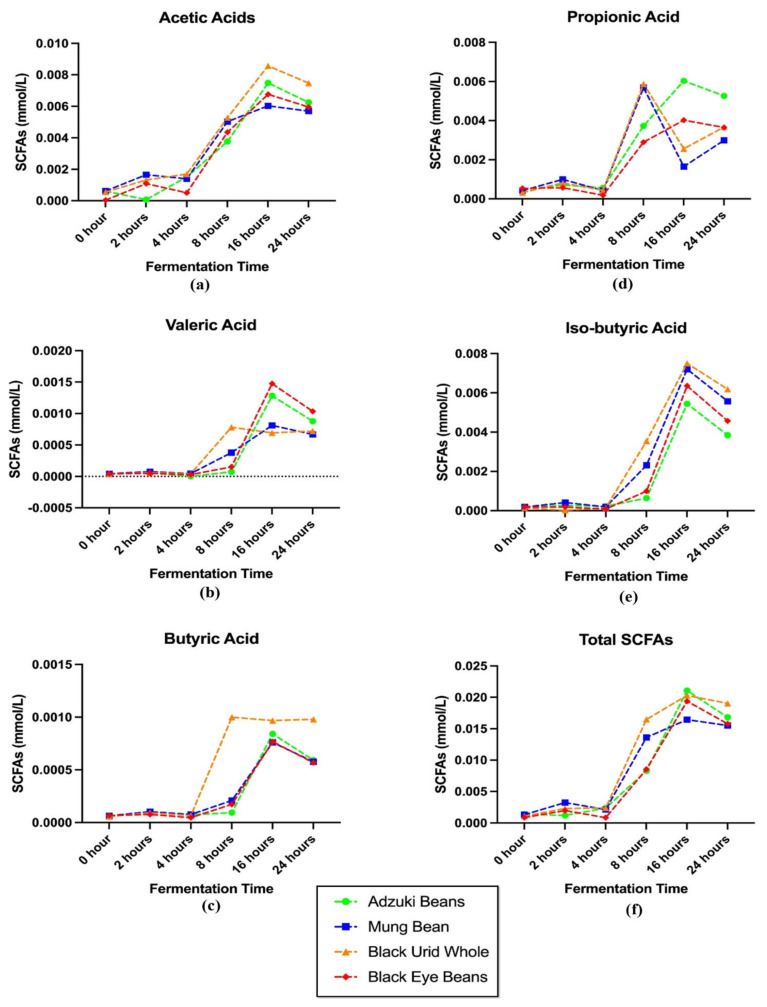
The trend of SCFAs production in different legumes digesta after complete colonic fermentation. (**a**) The level of acetic acid in legumes digesta; (**b**) the level of valeric acid in legumes digesta; (**c**) the level of butyric acid in legumes digesta; (**d**) the level of propionic acid in legumes digesta; (**e**) the level of iso-butyric acid in legumes digesta; (**f**) the level of total short chain fatty acids in legumes digesta.

**Table 1 foods-11-03884-t001:** Estimation of bioaccessibility (%) after complete in vitro digestion and colonic fermentation for phenolic compounds of legumes.

No.	Compound	Oral Bioaccessibility (%)				Gastric Bioaccessibility (%)				Intestinal Bioaccessibility (%)				Colonic Bioaccessibility (%)			
		AB *	MB	BUW	BEB	AB	MB	BUW	BEB	AB	MB	BUW	BEB	AB	MB	BUW	BEB
1	Gallic acid	179.12	99.30	84.82	76.32	91.91	98.81	84.63	75.91	92.11	98.77	84.75	76.52	97.02	108.96	94.57	81.18
2	Protocatechuic acid	-	100.61	0.00	-	-	100.70	0.00	-	-	101.15	0.00	-	-	103.83	103.29	-
3	Caftaric acid	94.51	100.11	-	-	99.46	0.00	-	-	95.00	0.00	-	-	0.00	0.00	-	-
4	*p*-hydroxybenzoic acid	123.82	121.26	100.23	0.00	0.00	0.00	0.00	0.00	0.00	0.00	0.00	0.00	0.00	0.00	0.00	100.13
5	Chlorogenic acid	102.46	-	-	101.82	101.35	-	-	100.41	101.20	-	-	100.70	101.14	-	-	0.00
6	Coumaric acid	-	98.87	-	-	-	0.00	-	-	-	0.00	-	-	-	0.00	-	-
7	Kaempferol	-	-	100.44	0.00	-	-	100.20	102.60	-	-	101.15	102.92	-	-	0.00	102.98
Total Phenolic compounds		157.25	179.85	106.48	79.20	102.35	32.07	66.40	130.75	100.76	32.14	66.94	102.09	167.54	84.95	62.90	96.76

* Legume samples mentioned in abbreviations are Adzuki Beans “AB”; Mung Bean “MB”, Black Urid Whole “BUW”, Black Eye Beans “BEB”. ‘-’ shows that compound not detected during HPLC analysis while “0.00” means bioaccessibility of compound is too low to quantify using equation.

**Table 2 foods-11-03884-t002:** Estimation of intestinal total recovery (%), colonic total recovery (%), residual intestinal digesta index (%) and residual colonic digesta index (%) after complete in vitro digestion and colonic fermentation for phenolic compounds of legumes.

No.	Compound	Intestinal Total Recovery (%)				Colonic Total Recovery (%)				Residual Intestinal Digesta Index (RID %)				Residual Colonic Digesta Index (RCD %)			
		AB *	MB	BUW	BEB	AB	MB	BUW	BEB	AB	MB	BUW	BEB	AB	MB	BUW	BEB
1	Gallic acid	191.54	206.89	177.41	159.08	188.82	208.35	183.00	158.52	99.43	108.12	92.65	82.56	91.79	99.38	88.43	77.33
2	Protocatechuic acid	-	207.74	106.36	-	-	103.83	204.02	-	-	106.58	106.36	-	-	0.00	100.73	-
3	Caftaric acid	95.00	0.00	-	-	95.50	0.00	-	-	0.00	0.00	-	-	95.50	0.00	-	-
4	*p*-hydroxybenzoic acid	0.00	0.00	0.00	0.00	0.00	0.00	0.00	100.13	0.00	0.00	0.00	0.00	0.00	0.00	0.00	0.00
5	Chlorogenic acid	101.20	-	-	200.91	101.14	-	-	102.12	0.00	-	-	100.21	0.00	-	-	102.12
6	Coumaric acid	-	0.00	-	-	-	0.00	-	-	-	0.00	-	-	-	0.00	-	-
7	Kaempferol	-	-	101.15	102.92	-	-	0.00	102.98	-	-	0.00	0.00	-	-	0.00	0.00
Total Phenolic compounds		213.68	170.25	146.02	183.85	216.51	99.31	87.22	128.38	112.92	138.11	79.08	81.76	48.97	14.36	24.32	31.62

* Legume samples mentioned in abbreviations are Adzuki Beans “AB”; Mung Bean “MB”, Black Urid Whole “BUW”, Black Eye Beans “BEB”. ‘-’ shows that compound not detected during HPLC analysis while “0.00” means bioaccessibility of compound is too low to quantify using equation.

## Data Availability

The data are available from the corresponding author.

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
