# Peer review of "In Vitro Gastrointestinal Bioaccessibility, Bioactivities and Colonic Fermentation of Phenolic Compounds in Different Vigna Beans"

_foods, 2022, doi:10.3390/foods11233884_

Round 1

Reviewer 1 Report

This study attempts to evaluate the phenolic compounds' antioxidant capacity as well as bioaccessability changes from four different Vigna beans. Metabolites formed were also measured.

I believe that this study is novel and the data reported adds significant knowledge in the area of food and function. The manuscript is well-written. The introduction and objectives were described clearly and most results were systematically presented. Discussions were based on sound evidence and conclusions reported the main findings. Well done.

Some minor mistakes and suggestions:

1. Method 2.6.2, reference [18] was written differently than other similar references.

2. The size of the bars on Figure 1 and Figure 2 should be increased to increase readability.

3. The conclusion should include some of the limitations of in vitro gastrointestinal digestion method compared to other methods. 

Reviewer 2 Report

I am very grateful you for the invitation to review the manuscript foods-2046450 by Li and coauthors "Simulated gastrointestinal digestion and in vitro colonic fermentation of Vigna beans: Bioaccessibility and bioactivities”. This research aimed to assess the distinctions in the content and bioaccessibility of phenolic compounds in four different species of Vigna samples (adzuki bean, black urid whole, black eye bean, and mung bean), their antioxidant properties and short-chain fatty acids (SCFAs) production through in vitro gastrointestinal digestion and colonic fermentation. The work is very interesting but needs a few adjustments to increase the quality of the material.

 Before presenting the few review comments, it is important to congratulate the authors for the design presented in the manuscript, in addition to the meticulous and detailed writing on all items.

Comments:

- Abstract: Please include the problem to be answered (nutrient supply, modification of bioactivities during digestion, etc).

- Keywords: Change the repeated keywords by different words from the title.

- Introduction: Insert information about the market, production and consumption of the species used.

- …“also on their bioaccessibility [8,9]”: Please insert the definition of bioaccessibility and factors that may interfere with the nutrient use. Thus, demonstrating the reason for the importance of evaluating.

- “They possess stronger protein contents, dietary fibers, and minerals than other main grains”: Include observed mean values.

- All these compounds may have various impacts on Vigna bioaccessibility: Specify main impacts and mechanisms.

- Introduction: A more in-depth explanation of the digestion of components should be presented, as it is the focus of the manuscript.

- Introduction: A more in-depth explanation of the digestion of components should be presented, as it is the focus of the manuscript.

- “was then utilized to grind the legumes into fine powder”: Specify the average granulometry.

- “the residue was alkaline and acidic hydrolyzed”: Solid residue?

- Standardize the use of min and minute units throughout the text.

- Methodology: It is very well written and detailed. A few remarks regarding the standardization of units are necessary.

- “as well as the minimal influence (1.25 μkat/mL) of α-amylase [20,21]” Highlight the role of the enzyme at this stage.

- 3.1.1. Phenolic Estimation: The authors discussed it properly and in detail. In general, in this or other items, it is necessary to highlight and discuss that generally the components are ingested in complex matrices and in this case, it was evaluated in isolation.

- 3.1.2. Antioxidant Activities Estimation: Authors must specify the groups or characteristics of the components detected and evaluated in each analysis.

- 3.2.1. Phenolic Estimation: At one point the authors discuss the role of microorganisms in the availability of components. It is interesting to add the effect of the components on the metabolic activity and even on the viability of microorganisms in the intestinal function.
